# Randomised Controlled Trials of Alcohol-Based Surgical Site Skin Preparation for the Prevention of Surgical Site Infections: Systematic Review and Meta-Analysis

**DOI:** 10.3390/jcm10040663

**Published:** 2021-02-09

**Authors:** Trisha N. Peel, Eliza Watson, Sue J. Lee

**Affiliations:** Department of Infectious Diseases, Alfred Hospital and Central Clinical School, Monash University, 85 Commercial Road, Melbourne, VIC 3004, Australia; eliza.watson@monash.edu (E.W.); sue.lee@monash.edu (S.J.L.)

**Keywords:** surgical site skin preparation, surgical site infection, infection prevention, meta-analysis

## Abstract

(1) Background: Surgical site skin preparation is an important approach to prevent postoperative wound infections. International guidelines recommend that alcohol-based combinations be used, however, the optimal combination remains uncertain. This study compares the effectiveness of alcohol-based chlorhexidine and alcohol-based iodophor for surgical site skin preparation for prevention of surgical site infections (SSIs). (2) Methods: Randomised controlled trials comparing alcohol-based interventions for surgical site skin preparation were included. The proportion of SSIs was compared using risk ratios (RR) with 95% confidence intervals (95% CI). The meta-analysis was performed with a fixed effect model using Mantel-Haenszel methods. As an a priori subgroup analysis SSI risk was examined according to different surgical procedural groups. (3) Results: Thirteen studies were included (*n* = 6023 participants). The use of chlorhexidine-alcohol was associated with a reduction in risk of SSIs compared with iodophor-alcohol (RR 0.790; 95% CI 0.669, 0.932). On sub-group analysis, chlorhexidine-alcohol was associated with a reduction in SSIs in caesarean surgery (RR 0.614; 95% CI 0.453, 0.831) however, chlorhexidine-alcohol was associated with an increased risk of SSI in bone and joint surgery (RR 2.667; 95% CI 1.051, 6.765). When excluding studies at high risk of bias on sensitivity analysis, this difference in alcohol-based combinations for bone and joint surgery was no longer observed (RR 2.636; 95% CI 0.995, 6.983). (4) Conclusions: The use of chlorhexidine-alcohol skin preparations was associated with a reduced risk of SSI compared to iodophor-alcohol agents. However, the efficacy of alcohol-based preparation agents may differ according to the surgical procedure group. This difference must be interpreted with caution given the low number of studies and potential for bias, however, it warrants further investigation into the potential biological and clinical validity of these findings.

## 1. Introduction

Over 300 million surgeries are performed annually worldwide based on a 2012 estimate [1]. Surgical site infections (SSIs) remain a major, costly complication of surgical procedures [2,3,4]. The patient’s skin bacteria is the major source of infecting pathogens involved in SSIs and is the target of infection prevention strategies such as surgical site skin preparation [5,6,7,8]. The three main agents used are chlorhexidine gluconate, iodophors or alcohol. Despite longstanding use, the optimal preparation remains an issue of controversy [9]. It is recommended that alcohol-based products, combining chlorhexidine with alcohol or iodophors with alcohol, be used in preference to aqueous-based products, based on improved efficacy demonstrated in randomised controlled trials and meta-analyses [6,7,8,9]. These agents have different mechanisms and duration of activity. Alcohol and chlorhexidine gluconate disrupt the cell wall of microorganisms, whereas iodophors act upon intracellular proteins of microorganisms [6,10]. Alcohol has no residual activity, iodophors exhibit persistence of bacteriostatic activity when on the skin and chlorhexidine gluconate has excellent residual activity [5,6,10].

In the guidelines for the prevention of SSIs published by The Centers for Disease Control and Prevention (CDC) [8] and the World Health Organization (WHO) [7,8], there was consensus that alcohol-based preparations were associated with reduced risk of SSI compared to aqueous-based solutions [7,8]. These guidelines, however, gave conflicting recommendations on the optimal agent to combine with alcohol. In the CDC guidelines, no recommendation for a specific alcohol-based product was made based on high-quality evidence from six randomised controlled trials [8]. In comparison, the WHO specified that chlorhexidine-alcohol preparation should be used, based on data from six randomised controlled trials of moderate quality [7]. The trials included a range of different procedure types and no sub-group analyses were performed to examine whether the efficacy of products differed between different surgery types.

The objective of this systematic review and meta-analysis was to determine the comparative effectiveness of alcohol-based chlorhexidine and alcohol-based iodophor as surgical site skin preparation agents to prevent SSIs based on data from randomized controlled trials. A secondary objective was to determine if the effectiveness of alcohol-based agents differs between different surgical procedure types.

## 2. Methods

This systematic review and meta-analysis was developed in keeping with the Preferred Reporting Items for Systematic Reviews and Meta-Analyses (PRISMA) Statement [11]. The intervention of interest was surgical site skin preparation with alcohol-based solutions or powders applied to the participant’s skin at the site overlying the planned surgical incision. The solutions could be applied as a single step (e.g., combination preparations such as 2% chlorhexidine-gluconate in 70% ethanol) or as two sequential steps (e.g., 10% povidone solution followed by 70% alcohol solution). Classes and sub-classes of surgical site skin preparation included, but were not limited to: chlorhexidine gluconate, povidone, iodine, alcohol and ethanol.

Randomised controlled trials (RCTs) in human participants that compared at least two of the alcohol-based interventions for surgical site skin preparation were eligible for inclusion. Studies that compared aqueous-based solutions to other aqueous-based solutions or to alcohol-based solutions were excluded. Studies comparing other antisepsis techniques (e.g., preoperative showering/bathing, impregnated drapes) were not included. Non-English language publications were excluded.

The primary outcome of interest was surgical site infection (SSI), based on the Centers for Disease Control and Prevention/National Healthcare Safety Network (CDC/NHSN) Surveillance definitions for SSIs [5,8]. In the event the trial did not apply the CDC definition, the study definition was documented and mapped to the CDC definition, where possible. Secondary outcomes of interest included: adverse events including skin irritation or allergic reactions and health economic data, including direct hospital costs, societal and quality of life data.

The following electronic databases were searched: Medline (via PubMed), OVID EMBASE, CINAHL and the Cochrane Library Databases. The strategy for electronic databases search is outlined in Appendix A. There was no time limit on the studies included. In addition, the reference list for included studies were reviewed for literature saturation. Study authors were not contacted, and the Grey Literature was not included. Database searches were completed August 2019.

Two authors (TP and EW) independently screened all titles and abstracts of identified studies using the Covidence^®^ (Veritas Health Innovation LTD, Melbourne, Australia) web-based platform. The full text was retrieved and reviewed for selected abstracts. Disagreement was resolved through consensus. Systematic reviews and meta-analyses were reviewed to identify additional randomised controlled trials. Data were extracted from the selected studies independently by two authors (TP and EW). Data extraction included the procedure type, number of participants, duration of follow up, the authors’ definition of surgical site infection, number of patients experiencing a surgical site infection and the interventions being compared. Arm level data were extracted. Risk of bias was assessed using a domain-based evaluation by two authors (TP and EW) independently [12]. Domains examined included random sequence generation, allocation concealment, blinding, attrition, and reporting biases. Disagreement was resolved through discussion and consensus. These data were extracted into RevMan^®^ (Review Manager [Computer program]. Version 5.3. Copenhagen: The Nordic Cochrane Centre, The Cochrane Collaboration, 2014). The protocol for the systematic review and meta-analysis was registered with PROSPERO (CRD42020148548).

The proportion of SSIs was compared between arms using risk ratios (RR) with 95% confidence intervals (95% CI). Statistical heterogeneity was quantified using the I^2^ statistic with a threshold at ≥50%. A negative I^2^ statistic was regarded as no heterogeneity. Funnel plots were examined for asymmetry applying the approach outlined by Sterne et al. [13]. Disclosed conflicts of interest were also reviewed.

The meta-analysis was performed with a fixed effect model using Mantel-Haenszel methods to obtain the pooled relative risk (RR) estimate [14,15]. In the event significant heterogeneity (I^2^ ≥ 50%) was observed, a random effects model using the restricted maximum likelihood method was performed. To account for trials with zero counts in either or both arms, a fixed continuity correction of 0.5 was added to each zero cell [16]. Sensitivity analyses were performed excluding trials with zero total events and, excluding trials with high risk of bias.

As part of the planned subgroup analysis the difference in risk of infection was examined according to different classifications of SSI (superficial and, deep or organ/space) and different surgical procedural groups. Data were analysed using STATA (16.0 College Station, TX, USA) with the META function.

## 3. Results

The initial search identified 4606 citations (Figure 1). Fifty-nine publications were subsequently retrieved for full-text review, of which 46 did not meet the inclusion criteria. Thirteen studies were eligible for inclusion in the review (*n* = 6023 participants) [17,18,19,20,21,22,23,24,25,26,27,28,29]. Characteristics of the included studies are outlined in Table 1.

All thirteen included trials compared alcohol-based chlorhexidine to alcohol-based iodophor preparations. The concentration of the chlorhexidine was 0.5% in six trials [17,19,20,23,24,29] and 2% in six trials [18,22,25,26,27,28]. In one trial [21], the concentration of the agents was not specified. The iodophor comparators included povidone-iodine (concentration range 1–10%) or free iodine (concentration range 0.7–1%). The alcohol concentration was documented in eight trials and ranged from 70–74%. In one trial [20], the iodophor and alcohol was applied in a two-step process. 

Overall, 3026 (50.2%) participants were randomised to chlorhexidine-alcohol preparations and 2997 (49.8%) participants to alcohol-iodophor preparations (Table 1). Participant follow-up ranged from 3 days to 365 days. Six of the studies applied the CDC criteria for defining SSI [18,20,21,23,27,29]. In three studies, infections were defined based on clinical diagnosis [17,24,28] and in four studies, no definitions for infection were provided [19,22,25,26]. Two trials [19,25] did not report any SSIs in either arm.

SSIs were reported in 7.1% (216/3026) participants in the chlorhexidine-alcohol group compared with 9.0% (271/2997) participants in the iodophor-alcohol group (Table 2). The use of chlorhexidine-alcohol was associated with a reduction in risk of SSI on pooled analysis (RR 0.790; 95% CI 0.669, 0.932: I^2^ 38.33% (95% CI 0 to 68.0%): and Figure 2). Results were similar when analysed without continuity correction (RR 0.782; 95% CI 0.662, 0.924: I^2^ 38.43% (95% CI 0–70.0)).

On planned sub-group analysis examining different procedure groups (Figure 3), chlorhexidine-alcohol was associated with a reduction in SSIs in caesarean surgery (RR 0.614; 95% CI 0.453, 0.831: I^2^ 39.81% (95% CI 0 to 81.4)). In contrast, the use of chlorhexidine-alcohol preparation was associated with an increased risk of SSI in bone and joint surgery (RR 2.667; 95% CI 1.051, 6.765: I^2^ 0.00% (95% CI 0 to 35.7)). In general surgery and skin and soft tissue surgery, there was no association with skin preparation and risk of SSI (RR 0.844; 95% CI 0.686, 1.038: I^2^ 49.55% (95% CI 0 to 85.4) and RR 0.460; 95% CI 0.105, 2.024: I^2^ 58.83% (95% CI 0 to 90.2), respectively).

Repeating the analysis without continuity correction did not alter the findings for caesarean, general or skin and soft tissue surgery (Caesarean Surgery RR 0.614, 95% CI 0.453, 0.831, General Surgery RR 0.844; 95% CI 0.686, 1.038 and, Skin and Soft Tissue Surgery RR 0.495; 95% CI 0.097, 2.517). While still at increased risk, using chlorhexidine-alcohol preparation for bone and joint surgery was no longer statistically significant (RR 3.000; 95% CI 0.995, 9.045). 

Examining the impact of alcohol-based surgical site skin preparation according to SSI classification did not reveal any significant associations between different alcohol-based surgical skin site preparation agents for superficial SSI (*n* = 12 trials, RR 0.807; 95% CI 0.632, 1.032: I^2^ = 0.00% (95% CI 0 to 71.0)) or deep and organ/space SSI (*n* = 12 trials, RR 0.904; 95% CI 0.664, 1.230: I^2^ = 0.00% (95% CI 0 to 54.9)) (Table 2).

Adverse events, including allergic reactions to the preparation, were reported in three studies [18,23,27] (Table 1 and Table 2). Adverse reactions were rare, occurring in 0.15% of participants allocated to chlorhexidine-alcohol (2/1354) and 0.29% allocated to iodophor-alcohol (4/1361, *p* = 0.687). Tuuli et al. [27] reported on healthcare resource utilisation including emergency room visits and did not find any difference between chlorhexidine-alcohol and iodine-alcohol. No other study reported on health economic outcomes.

The risk of bias was assessed as low in two studies [23,27] (Figure 4A). In nine trials, the risk of bias was unclear, particularly the risk of performance and detection bias [17,18,19,20,21,25,26,28,29]. There was high risk of selection bias determined in the trials by Rodrigues et al. [24] and Ostrander et al. [22] (Figure 4A). Overall, the risk of reporting bias and attrition bias was low (Figure 4B). Excluding the trials with high risk of bias did not alter the overall estimates (*n* = 11 trials, RR 0.765; 95% CI 0.646, 0.907; *p* = 0.0020: I^2^ = 39.72% (95% CI 0 to 70.3)), however, on sub-group analysis of different procedure groups, when the trial by Ostrander et al. was excluded, chlorhexidine-alcohol preparation for bone and joint surgery was no longer statistically significant (RR 2.636; 95% CI 0.995, 6.983; I^2^ = 0.00% (95% CI 0 to 66.1)).

The funnel plot (Figure 5A) was asymmetric. When repeated according to procedure group, the plots were symmetric (Figure 5B) although the number of the trials was small, thereby limiting analysis for bias.

## 4. Discussion

Overall, our study, including 6023 participants, showed the use of chlorhexidine-alcohol skin preparations was associated with a 21% reduction in the relative risk of SSI compared to iodophor-alcohol agents. This equates to an absolute difference of 19 fewer infections per 1000 patients undergoing surgery. A key finding however, suggests that the efficacy of alcohol-based surgical site skin preparation agents may differ according to the surgical procedure group. In caesarean surgery, the use of chlorhexidine-alcohol preparations was associated with a 39% reduction in the relative risk of SSIs. In contrast, chlorhexidine-alcohol skin preparation agents were associated with a 2.7-fold increased risk of SSIs in bone and joint surgery. In the other procedure groups, general surgery and skin and soft tissue surgery, there was no demonstrated difference between the alcohol-based agents.

The observation in the bone and joint surgery cohort differs from the WHO guidelines which recommended that alcohol-based chlorhexidine solutions should be used. The meta-analysis performed by the WHO Guideline writing group include three studies examining bone and joint surgery [19,25,26]. The estimates in bone and joint surgery were influenced by one large study by Peel et al. [23] although examination of the forest plot suggests the majority of trials in this surgical group favoured alcohol-based iodophor skin preparation agents. The larger study by Peel et al. (*n* = 780) [23] examined skin preparation agents in arthroplasty surgery and was reported after the WHO guidelines were published.

The findings in caesarean surgery were influenced by the trials by Tuuli et al. [27] and Kesani et al. [20]. Of note, the trial by Tuuli et al. was published outside systematic review time limits for the WHO meta-analysis however, was “exceptionally included” [3,7]. This trial also was published after the specified time limits for the CDC meta-analysis [8] and was not included in the meta-analysis, potentially accounting for the differing findings between the two guidelines.

The differences observed, particularly between bone and joint surgery and caesarean surgery, may be due to differences in the populations, particularly age and gender. It may also reflect the differences in infection control approaches, for example, screening and decolonisation for *Staphylococcus aureus* is a recommended strategy in orthopaedic surgery whereas the role has not been established in caesarean surgery [7,8]. The causative organisms of SSI also differ between these surgical groups: *Staphylococcus* species are the most common bacteria isolated in bone and joint infections, compared with Gram negative bacteria and anaerobes in caesarean surgery [30,31]. The majority of the cohort in the bone and joint surgery group underwent procedures involving the implantation of prosthetic material. Infections involving prosthetic material differ from other types of SSIs due to the propensity of organisms involved in medical device infections to form biofilm [8]. This observation raises the possibility that the differences between bone and joint and caesarean surgery may relate to different anti-biofilm properties of the skin preparation agents. Both chlorhexidine and iodophors display anti-biofilm properties [32,33,34]. The effectiveness of the agents forming biofilms may be concentration dependent [34,35]. In a trial by Smith and colleagues, lower concentrations of chlorhexidine were less effective at eradicating methicillin resistant *Staphylococcus aureus* biofilm compared to 2 and 4% concentrations [35]. The trial by Peel et al. used 0.5% chlorhexidine however, conversely there were more cases of *Staphylococcus aureus* SSI in the iodophor-alcohol arm (3/4 SSIs: 75%) compared with the chlorhexidine-alcohol arm (3/12 SSIs: 25%) [23].

Given that this meta-analysis included a small number of trials (*n* = 13), the observed difference may be a false positive finding, due to chance. Particularly for the analysis according to procedure group, the estimates must be interpreted with caution given the small number of trials included in each sub-group.

The trials included in this study compared different concentrations and formulations of skin preparation agents. The impact on the estimates of differing concentrations is unclear. There is limited data on the optimal concentration of agents, with no head-to-head trials comparing concentrations [3,7,9,36]. In addition, a range of definitions for SSI were applied in the included studies and participant follow-up differed. The heterogeneity, however, was low. Finally, the majority of trials included had unclear risk of bias, particularly for performance and detection biases.

## 5. Conclusions

The results of this meta-analysis suggest that the use of alcohol-based chlorhexidine surgical site skin preparation for caesarean section is associated with a lower risk of surgical site infections compared with alcohol-based iodophors. The opposite finding was observed for bone and joint surgical procedures, raising the possibility that the optimal skin preparation agent may differ with surgical procedures. However, when excluding studies at risk of major bias, this difference for bone and joint procedures was no longer significant. These observations must be interpreted with caution and require further investigation to corroborate these findings and to determine if there is a biological mechanism(s) explaining these findings. Further, larger trials, particularly in other surgical procedure groups, are warranted.

## Figures and Tables

**Figure 1 jcm-10-00663-f001:**
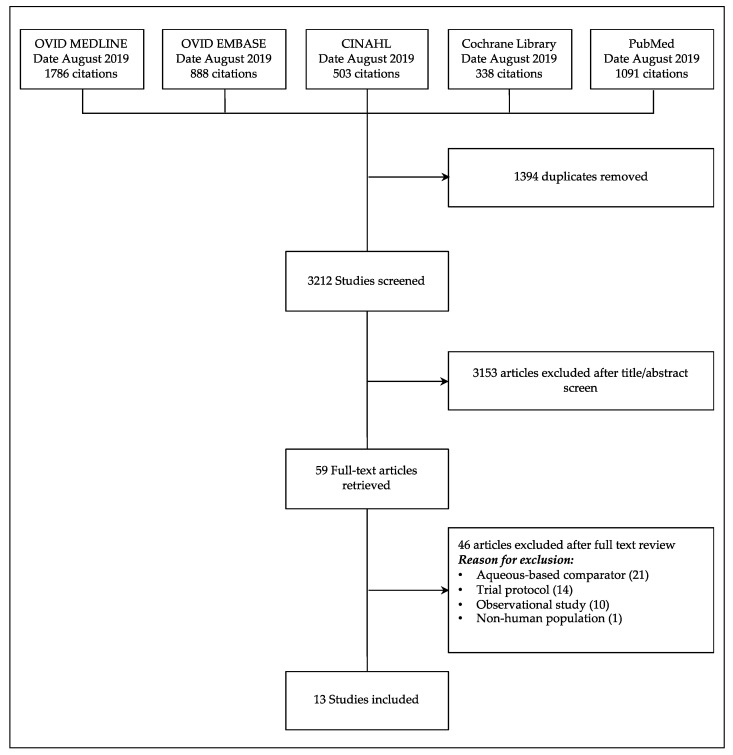
PRISMA Flow Chart of Systematic Review and Study Selection.

**Figure 2 jcm-10-00663-f002:**
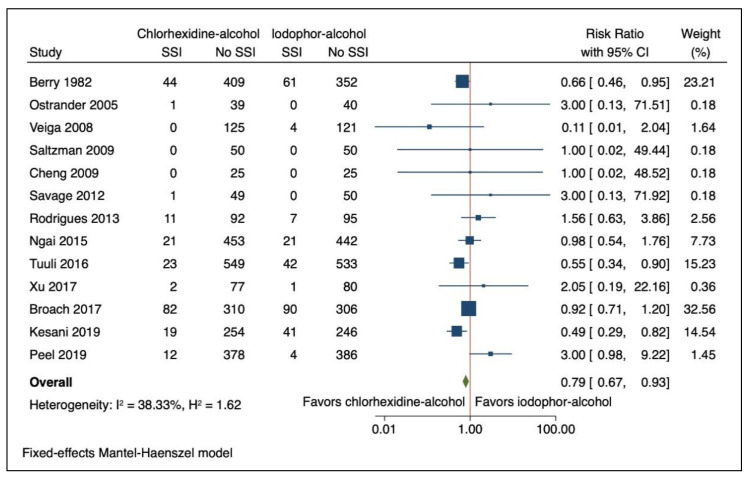
Forest Plot Comparing Risk of Surgical Site Infection (SSI) with Chlorhexidine-Alcohol to Iodophor-Alcohol Surgical Site Skin Preparation. Navy square represents the effect estimates from the individual studies, the size of the square is proportional to the weight of the study. The horizontal line represents the 95% confidence interval of the study estimate. The green diamond represents the pooled effect size.

**Figure 3 jcm-10-00663-f003:**
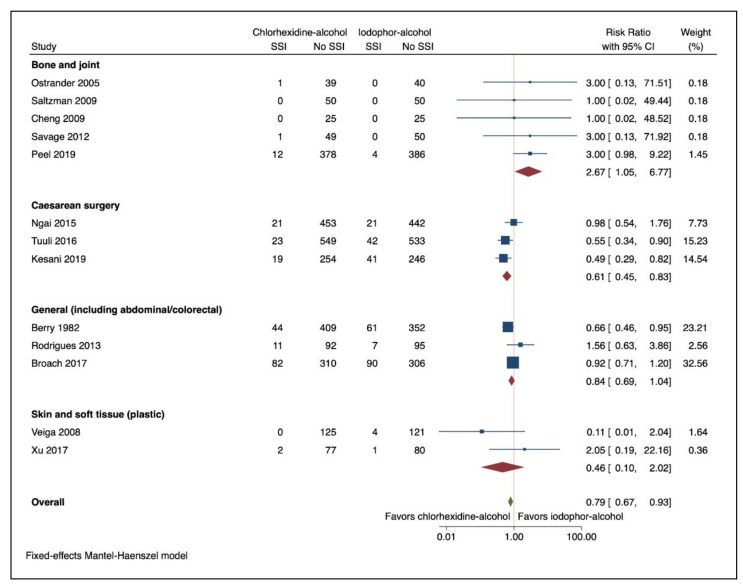
Forest Plot Comparing Risk of Surgical Site Infection (SSI) with Chlorhexidine-Alcohol to Iodophor-Alcohol Surgical Site Skin Preparation According to Procedure Groups. Navy square represents the effect estimates from the individual studies, the size of the square is proportional to the weight of the study. The horizontal line represents the 95% confidence interval of the study estimate. The red diamond represents the pooled effect size for the specified procedure group. The green diamond represents the pooled effect size.

**Figure 4 jcm-10-00663-f004:**
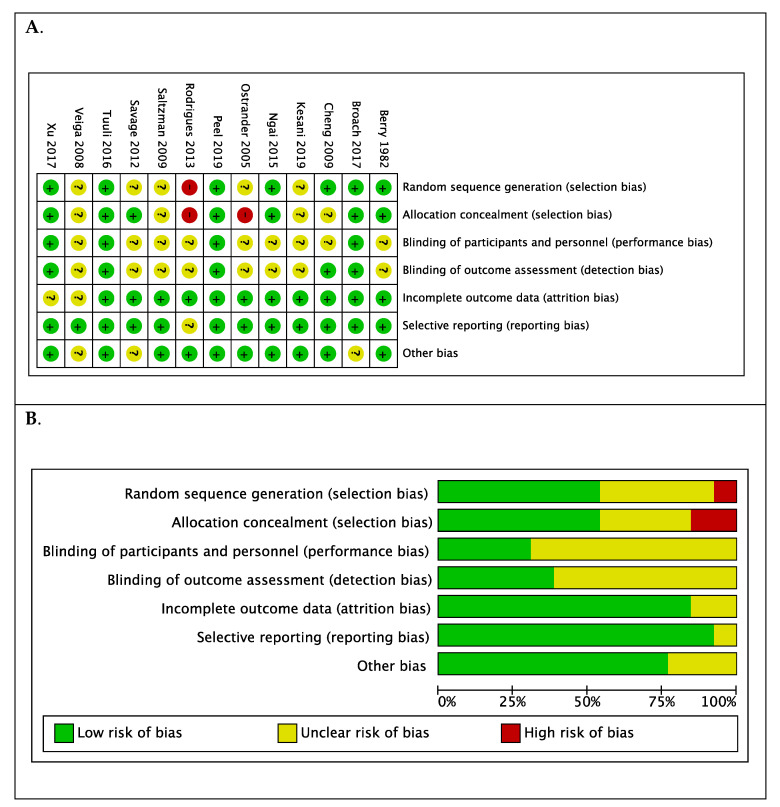
(**A**). Summary of risk of bias for each trial: low (+), high (−), or unclear (?) and (**B**). Each risk of bias item presented as percentages across all included studies.

**Figure 5 jcm-10-00663-f005:**
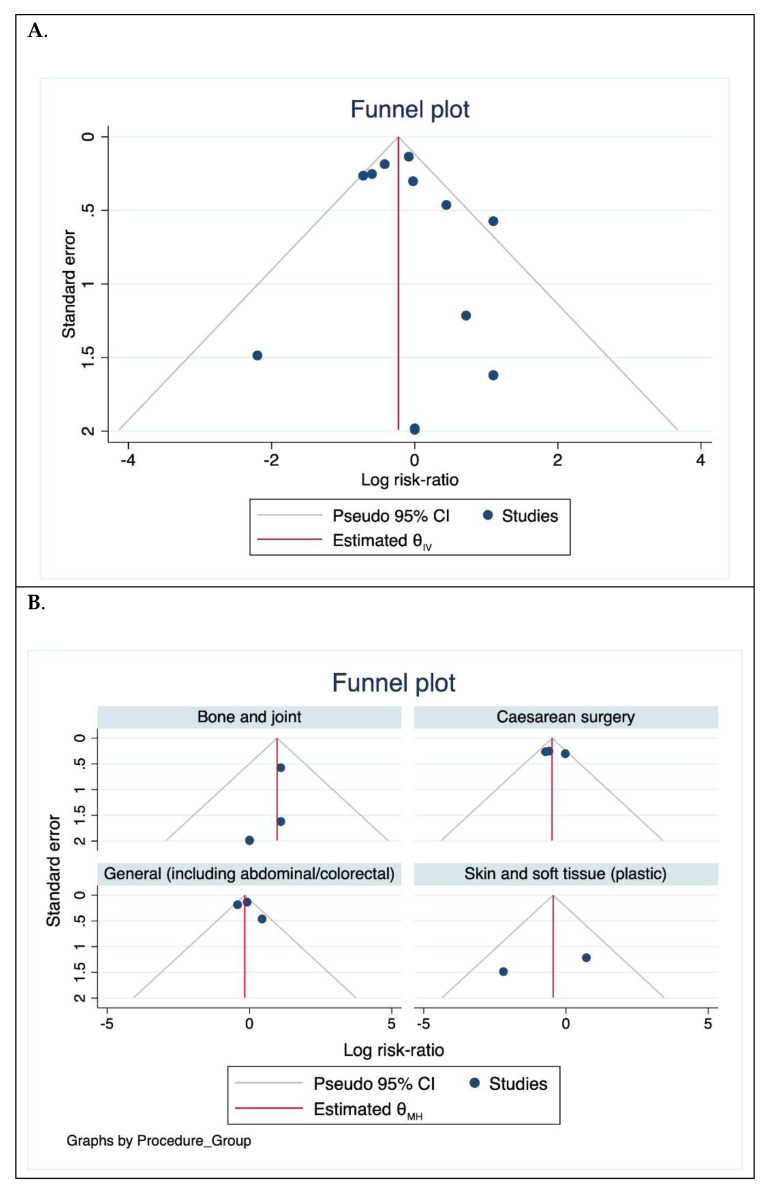
Funnel plot assessing publication bias. (**A**). For all trials included and (**B**). According to procedure groups.

**Table 1 jcm-10-00663-t001:** Characteristics of Included Studies.

Study	Methods	Surgery Type	Interventions	Outcomes	Results	Notes
Berry 1982 [17]	Cluster randomised controlled trial Clustered by given day of surgery Single centre study	Operations on biliary tract, large bowel, laparotomy, operations on hernia, genitalia, varicose veins and other ‘clean’ non-abdominal operations	Group A Povidone-iodine 10%in alcohol (*n* total = 413; included in trial analysis *n* = 413) Group B Chlorhexidine (‘Hibitane’) 0.5% in spirit (*n* total = 453; included in trial analysis *n* = 453*)*	Wound infection defined as any wound abnormality at time of participant’s discharge as agreed by two observers. Wounds were judged at each inspection as fitting one or more of the following categories: normal, erythematous, oedematous, discharging or purulent.	SSI Iodophor-alcohol 61/413 (14.8%) Chlorhexidine-alcohol 44/453 (9.7%)	No sample size estimate provided Follow up: until discharge Data extracted from wound abnormalities at discharge (Table III in the publication by Berry et al.) based on participants with “any abnormality” No data provided on whether wound abnormalities were superficial, deep or organ/space
Broach 2017 [18]	Randomised controlled trial Blinded Non-inferiority trial design Setting of study is not stated	Elective clean-contaminated colorectal surgery	Group A 26 mL single use applicator containing iodine povacrylex [0.7% available iodine]/74% isopropyl alcohol (*w/w*) [“Duraprep”] (*n* total = 402; included in trial analysis *n* = 396*)* Group B 26 mL single-use applicator containing 2% chlorhexidine gluconate (*w/v*) and 70% isopropyl alcohol (*v/v*) [“ChloraPrep”]. (*n* total = 400; included in trial analysis *n* = 392*)*	Surgical site infection at 30 days post discharge (±5 days) applying the CDC definition of superficial or deep surgical site infection. Cellulitis and organ/space SSI included as a secondary outcome measure	SSI All SSI Iodophor-alcohol 90/396 (22.7%) Chlorhexidine-alcohol 82/392 (20.9%) Superficial SSI Iodophor-alcohol 46/396 (11.6%) Chlorhexidine-alcohol 40/392 (10.2%) Deep SSI Iodophor-alcohol 28/396 (7.1%) Chlorhexidine-alcohol 22/392 (5.6%) Organ space SSI Iodophor-alcohol 16/396 (4.0%) Chlorhexidine-alcohol 20/392 (5.1%) Cellulitis Iodophor-alcohol 19/396 (4.8%) Chlorhexidine-alcohol 14/392 (3.6%) Adverse reactions No adverse reactions (skin irritation/allergy) reported in either arm	Sample size estimate provided For the analysis, organ space SSI included with data from primary outcome (superficial and deep SSI) Follow-up: 30 days ± 5 days
Cheng 2009 [19]	Randomised controlled trial Single centre study	Foot surgery including: metatarsal osteotomies for correction of hallux valgus deformity, removal of osteophytes from the first metatarsal and correction of lesser toe deformities	Group A alcoholic betadine (Ecolab Videne Alcoholic tincture, povidine-iodine 10% *w/w* (1% *w/w* available iodine)) (*n* total = 25; included in trial analysis *n* = 25) Group B alcoholic chlorhexidine (Ecolab Hydrex, clear Chlorhexidine gluconate 0.5% *w/v* in 70% *v/v*) (*n* total = 25; included in trial analysis *n* = 25)	Number of bacterial colony forming units pre- and post-treatment with surgical site skin preparation taken from three sites (medial hallucal nail fold, interdigital web-spaces and the dorsal aspect of the first metatarsal phalangeal joint). Postoperative infection rate was reported as a secondary outcome. No definition for infection provided	Positive cultures Iodine-alcohol: positive cultures prior to preparation 62, after 9 from 3 sites in 25 participants Chlorhexidine-alcohol: positive cultures prior to preparation 55, after 4 from 3 sites in 25 participants Postoperative infection No participant allocated to either iodine-alcohol or chlorhexidine developed a post-operative infection	In addition to the skin preparation on the surgical site (the foot), the participant’s opposite foot (i.e., the foot not being operated upon) was scrubbed with a bristle brush for 3-min before the same preparation was applied. For the purposes of analysis, only data from the surgical site included. Sample size estimate provided Follow-up: not defined
Kesani 2019 [20]	Randomised controlled trial Single centre study	Caesarean section	Group A povidone iodine-alcohol (10% povidone-iodine and then with surgical spirit) (*n* total = 287; included in trial analysis *n =* 287) Group B chlorhexidine alcohol (2% chlorhexidine gluconate and 70% isopropyl alcohol) (*n* total = 273; included in trial analysis *n* = 273)	SSI defined as per the CDC definitions. Superficial and deep SSI reported.	SSI All SSI Iodophor-alcohol 41/287 (14.3%) Chlorhexidine-alcohol 19/273 (7.0%) Superficial SSI Iodophor-alcohol 29/287 (10.1%) Chlorhexidine-alcohol 15/273 (5.5%) Deep SSI Iodophor-alcohol 12/287 (4.2%) Chlorhexidine-alcohol 4/273 (1.5%) No difference in microorganisms from cultured reported.	Sample size estimate not provided Follow-up: 30 days postoperatively
Ngai 2015 [21]	Randomised controlled trial Three-arm study Two centres	Caesarean section	Group A povidone iodine-alcohol (concentrations not stated) (*n* total = 463; included in trial analysis *n* = 463) Group B chlorhexidine alcohol (concentrations not stated) (*n* total = 474; included in trial analysis *n* = 474) Group C combination of povidone iodine-alcohol (concentrations not stated) applied first followed by chlorhexidine alcohol (concentrations not stated) (*n total = 467; included in trial analysis n = 467)*	Surgical site infection defined according to the CDC definition assessed at 2- and 6-weeks post-cesarean	SSI All SSI Iodophor-alcohol 21/463 (4.5%) Chlorhexidine-alcohol 21/474 (4.4%) Combination 18/467 (3.9%) Superficial SSI Iodophor-alcohol 16/463 (3.5%) Chlorhexidine-alcohol 15/474 (3.2%) Combination 15/467 (3.2%) Deep SSI Iodophor-alcohol 3/463 (0.6%) Chlorhexidine-alcohol 3/474 (0.4%) Combination 1/467 (0.2%) Organ space Iodophor-alcohol 2/463 (0.4%) Chlorhexidine-alcohol 3/474 (0.6%) Combination 2/467 (0.4%)	Analysis included data for participants administered povidone iodine-alcohol or chlorhexidine alcohol arms. Data from the combination arm with povidone iodine-alcohol followed by chlorhexidine alcohol excluded from analysis Sample size estimate provided Follow-up: 30 days postoperatively
Ostrander 2005 [22]	Randomised controlled trial Three-arm study Single centre study	Foot and ankle surgery	Group A “DuraPrep” [0.7% available iodine/74% isopropyl alcohol (*w/w*)] (*n* total = 40; included in trial analysis *n* = 40) Group B “Techni-Care” [3.0% chloroxylenol] (*n* total = 40; included in trial analysis *n* = 40) Group C “ChloraPrep” [2% chlorhexidine gluconate (*w/v*) and 70% isopropyl alcohol (*v/v*)]. (*n* total = 40; included in trial analysis *n* = 40)	Number of bacterial colonies forming units pre- and post-treatment with surgical site skin preparation: specimens collected from three sites: tibia, 12 cm proximal to the ankle Joint (labelled as control), hallucal nail fold (labelled as hallux), web spaces between the second and third and between the fourth and fifth digits (labelled as toe site). Postoperative infection rate was reported as a secondary outcome. No definition for infection provided	Positive cultures Iodophor-alcohol: 65% of hallux cultures positive, 45% of toe cultures and 23% of control cultures positive in 40 participants Chlorhexidine-alcohol: 30% of hallux cultures positive, 23% of toe cultures and 10% of control cultures positive in 40 participants Chloroxylenol: 95% of hallux cultures positive, 98% of toe cultures and 35% of control cultures positive in 40 participants Postoperative infections Iodophor-alcohol 0/40 (0%) Chlorhexidine-alcohol 1/40 (2.5%) Chloroxylenol: 2/40 (5.0%)	All procedures performed by one surgeon Analysis included data for participants administered DuraPrep [0.7% available iodine/74% isopropyl alcohol] or ChloraPrep [2% chlorhexidine gluconate/70% isopropyl alcohol] arms. Data from the Techni-Care (3.0% chloroxylenol) arm excluded from analysis. Sample size not met Follow-up: not defined
Peel 2019 [23]	Cluster randomised controlled trial Clustered by given day of surgery Single centre study	Elective hip or knee arthroplasty	Group A 1% iodine (*w/v*) in 70% ethanol (*v/v*) (*n* total = 390; included in trial analysis *n* = 390) Group B 0.5% chlorhexidine gluconate (*w/v*) in 70% ethanol (*v/v*) (*n* total = 390; included in trial analysis *n* = 390)	Primary outcome: superficial wound complication (composite of superficial incisional SSI and/or clinically significant wound ooze) Secondary outcome: SSI (superficial incisional SSI and/or prosthetic joint infection (PJI) based on the CDC definition	SSI Superficial and prosthetic joint infection (PJI) Iodophor-alcohol 4/390 (1.0%) Chlorhexidine-alcohol 12/390 (3.1%) Superficial SSI Iodophor-alcohol 3/390 (0.9%) Chlorhexidine-alcohol 4/390 (1.1%) Prosthetic joint infection Iodophor-alcohol 2/390 (0.6%) Chlorhexidine-alcohol 7/390 (2.0%) Superficial wound complication (composite of wound ooze and superficial SSI) Iodophor-alcohol 15/390 (3.8%) Chlorhexidine-alcohol 19/390 (4.9%) Wound Ooze Iodophor-alcohol 13/390 (3.3%) Chlorhexidine-alcohol 15/390 (3.8%) Adverse reactions No adverse reactions (skin irritation/allergy) reported in either arm	Sample size provided Follow-up: 30 days for primary outcome and 365 days for secondary outcome 90 Protocol violations observed. Sensitivity analysis with per-protocol and as-treated analysis
Rodrigues 2013 [24]	Randomised controlled trial Single centre study	Abdominal and thoracic surgery with subcostal abdominal, vertical abdominal and thoracic incision.	Group A 10% hydroalcoholic povidone-iodine (*n* total = 102; included in trial analysis *n* = 102) Group B 0.5% alcoholic chlorhexidine (*n* total = 103; included in trial analysis *n* = 103)	Diagnosis for SSI based on clinical, microbiological and radiological findings. Definition maps to the CDC definition for SSI. Diagnosis of SSI required at least one of the following signs: fever, without other apparent cause, pain, heat, swelling, or confluent erythema around the incision and extrapolating the boundaries of the wound, pus in the incision site or in the deep soft tissue, or in organ/cavity handled during operation; presence of abscesses or, in the case of deep tissues, histological or radiological evidence suggestive of infection; isolated microorganism from theoretically sterile source or harvested with aseptic technique from a previously closed site, and spontaneous dehiscence of deep tissues.	SSI All SSI Iodophor-alcohol 7/102 (6.7%) Chlorhexidine-alcohol 11/103 (10/7%) Superficial SSI Iodophor-alcohol 5/102 (4.9%) Chlorhexidine-alcohol 9/103 (8.7%) Deep SSI Iodophor-alcohol 1/102 (1.0%) Chlorhexidine-alcohol 2/103 (1.9%) Organ space Iodophor-alcohol 1/102 (1.0%) Chlorhexidine-alcohol 0/103 (0%)	Sample size not provided Follow-up: 30 days
Saltzman 2009 [25]	Randomised controlled trial Three-arm study Single centre study	Shoulder surgery including shoulder arthroplasty (*n* = 4)	Group A “DuraPrep” [0.7% iodophor and 74% isopropyl alcohol] (*n* total = 50; included in trial analysis *n* = 50) Group B “ChloraPrep” [2% chlorhexidine gluconate and 70% isopropyl alcohol]. (*n* total = 50; included in trial analysis *n* = 50) Group C povidone-iodine scrub and paint [0.75% iodine scrub and 1.0% iodine paint] (*n* total = 50; included in trial analysis *n* = 50)	Rate of positive microbiological cultures pre- and post-treatment with surgical site skin preparation. Postoperative infection rate was reported as a secondary outcome. No definition for infection provided	Positive cultures Iodophor-alcohol: 19% of cultures were positive in 50 participants Chlorhexidine-alcohol: 7% of cultures were positive in 50 participants Aqueous-based iodophor: 31% of cultures were positive in 50 participants Postoperative infections No postoperative infections developed in any intervention arm	All surgery performed by three surgeons Analysis included data for participants administered DuraPrep or ChloraPrep arms. Data from the povidone-iodine scrub and paint arm excluded from analysis. Sample size provided Follow-up: 10 months minimum
Savage 2012 [26]	Randomised controlled trial Single centre study	Lumbar spine surgery, including: microdisectomy, posterior spinal fusion with or without an associated interbody fusion decompression, kyphoplasty	Group A “DuraPrep” [0.7% iodophor and 74% isopropyl alcohol] (*n* total = 50; included in trial analysis *n* = 50) Group B “ChloraPrep” [2% chlorhexidine gluconate and 70% isopropyl alcohol]. (*n* total = 50; included in trial analysis *n* = 50)	Rate of positive microbiological cultures pre- and post-treatment with surgical site skin preparation. Postoperative infection rate was reported as a secondary outcome. No definition for infection provided	Positive cultures Iodophor-alcohol: 80% of cultures positive pre-preparation, 6% positive post-preparation, 32% positive post-closure in 50 participants Chlorhexidine-alcohol: 84% of cultures positive pre-preparation, 0% positive post-preparation, 34% positive post-closure in 50 participants Postoperative infection Iodophor-alcohol 0/50 (0%) Chlorhexidine-alcohol 1/50 (2.0%): All classed as superficial	All surgery performed by four surgeons Sample size provided Follow-up: 6 months minimum
Tuuli 2016 [27]	Randomised controlled trial Single centre study	Caesarean section	Group A chlorhexidine–alcohol combination [2% chlorhexidine gluconate with 70% isopropyl alcohol] (*n* total = 575; included in trial analysis *n =* 575) Group B iodine–alcohol combination [8.3% povidone–iodine with 72.5% isopropyl alcohol] (*n* total = 572; included in trial analysis *n* = 572)	Superficial or deep SSI based on the CDC/National Healthcare Safety Network definitions	SSI All SSI Iodophor-alcohol 42/575 (7.3%) Chlorhexidine-alcohol 23/572 (4.0%) Superficial SSI Iodophor-alcohol 28/575 (4.9%) Chlorhexidine-alcohol 17/572 (3.0%) Deep SSI Iodophor-alcohol 14/575 (2.4%) Chlorhexidine-alcohol 6/572 (1.0%) Adverse reactions Iodophor-alcohol 4/575 (0.7%) Chlorhexidine-alcohol 2/572 (0.3%) In a post hoc analysis, the use of healthcare resources did not differ intervention arms	Sample size provided Follow-up: 30 days
Veiga 2008 [29]	Randomised controlled trial Single centre study	Elective and clean plastic surgery procedures including: breast surgery, abdominoplasty, scar revision, zetaplasty, lipoma excision	Group A alcohol solution of povidone- iodine 10% (*n* total = 125; included in trial analysis *n =* 125) Group B alcohol solution of chlorhexidine 0.5% (*n* total = 125; included in trial analysis *n =* 125)	Primary outcome was quantitative skin cultures before and after surgical site skin preparation. Surgical site infection was a secondary outcome, defined according to the CDC definition	Colony-forming units (standard deviation) Iodophor-alcohol: 75.4 (115.9) colony-forming units pre-preparation, 1.3 (5.7) colony-forming units 2-min after preparation, 17.6 (64.7) colony-forming units at end of surgery in 125 participants Chlorhexidine-alcohol: 93.8 (127.3) colony-forming units pre-preparation, 0.3 (1.3) colony-forming units 2-min after preparation, 7.8 (46.1) colony-forming units at end of surgery in 125 participants SSI Postoperative infection Iodophor-alcohol 4/125 (03.2%) Chlorhexidine-alcohol 0/125 (0%) All classed as superficial	Study authors reported four infections (in 125 participants) in the iodophor-alcohol intervention arm calculated as 3.2%. The study authors however reported this proportion as “1.6%” (equating to 2 infections in 125 participants). Given this discrepancy, for the analysis, the number of events was transcribed as 4 infections. Sample size not provided Follow-up: 30 days
Xu 2017 [28]	Randomised controlled trial Three arm trial Single centre study	Elective clean soft tissue hand surgery (e.g., carpal tunnel release, trigger finger, de Quervain release, mass excision or excision ganglion cyst)	Group A “DuraPrep” [0.7% iodophor and 74% isopropyl alcohol] (*n* total = 81; included in trial analysis *n* = 81) Group B “ChloraPrep” [2% chlorhexidine gluconate and 70% isopropyl alcohol]. (*n* total = 79; included in trial analysis *n* = 79) Group C “Betadine solution” [10% povidone-iodine] (*n* total = 80; included in trial analysis *n =* 80)	Rate of positive microbiological cultures pre- and post-treatment with surgical site skin preparation. Postoperative infection rate was reported as a secondary outcome. Postoperative infection defined as defined as need for antibiotics or surgical intervention.	Positive cultures Iodophor-alcohol: cultures positive in 24 of 81 participants (29.6%) pre-preparation and 3 of 81 participants (3.7%) post-preparation Chlorhexidine-alcohol: cultures positive in 32 of 79 participants (40.5%) pre-preparation and 21 of 79 participants (26.6%) post-preparation Aqueous-iodophor: cultures positive in 35 of 80 participants (43.8%) pre-preparation and 1 of 80 participants (1.3%) post-preparation SSI Postoperative infections Iodophor-alcohol 1/81 (1.2%) Chlorhexidine-alcohol 2/79 (2.5%) Aqueous-iodophor 1/80 (1.3%) All classed as superficial	Number of participants allocated to intervention arms differs across Figure 1, Table 2 and in text. For the purposes of the analysis, the number of participants has been transcribed keeping with Figure 1 (Consort flow diagram). Analysis included data for participants administered DuraPrep or ChloraPrep arms. Data from the Betadine solution arm excluded from analysis. Sample size provided Follow-up: 42 days

**Table 2 jcm-10-00663-t002:** Summary of Effects Table.

**Patients or Population:** Patients undergoing surgery **Settings:** Hospital **Intervention:** Alcohol-based chlorhexidine surgical site skin preparation **Comparison:** Alcohol-based iodophor surgical site skin preparation					
**Outcomes**	**Absolute Effect**		**Absolute Risk Difference**	**Relative Effect * (95% CI)**	**Participants (Studies)**
	**Chlorhexidine-Alcohol**	**Iodophor-Alcohol**			
**All procedures**					
All SSI	216/3026 (7.1%)	271/2997 (9.0%)	There were 19 more SSIs per 1000 participants with iodophor-alcohol surgical site skin preparation compared with chlorhexidine-alcohol	RR 0.790 (0.669 to 0.932)	6023 (13 studies)
Superficial SSI	105/2573 (4.1%)	132/2584 (5.1%)	There were 10 more superficial SSIs per 1000 participants with iodophor-alcohol surgical site skin preparation compared with chlorhexidine-alcohol	RR 0.807 (0.632 to 1.032)	5157 (12 studies)
Deep or Organ Space SSI	67/2573 (2.6%)	79/2584 (3.1%)	There were 5 more deep or organ space SSIs per 1000 participants with iodophor-alcohol surgical site skin preparation compared with chlorhexidine-alcohol	RR 0.904 (0.664 to 1.230)	5157 (12 studies)
Adverse events	2/1354 (0.15%)	4/1361 (0.29%)	There is 1 more adverse event per 1000 participants with iodophor-alcohol surgical site skin preparation compared with chlorhexidine-alcohol	RR 0.603 (0.145 to 2.517)	2715 (3 studies)
**Bone and Joint Surgery**					
All SSI	14/555 (2.5%)	4/555 (0.7%)	There were 18 more SSIs per 1000 participants with chlorhexidine-alcohol surgical site skin preparation compared with iodophor-alcohol	RR 2.667 (1.051 to 6.765)	1110 (5 studies)
Superficial SSI	7/555 (1.3%)	3/555 (0.5%)	There were 8 more superficial SSIs per 1000 participants with chlorhexidine-alcohol surgical site skin preparation compared with iodophor-alcohol	RR 1.800 (0.607 to 5.335)	1110 (5 studies)
Deep or Organ Space SSI	7/555 (1.3%)	2/555 (0.4%)	There were 9 more deep or organ space SSIs per 1000 participants with chlorhexidine-alcohol surgical site skin preparation compared with iodophor-alcohol	RR 2.250 (0.697 to 7.266)	1110 (5 studies)
Adverse events	0/390 (0.0%)	0/390 (0.0%)	There were no events reported	RR 1.000 (0.020 to 50.271)	1110 (5 studies)
**Caesarean Surgery**					
All SSI	63/1319 (4.8%)	104/1325 (7.8%)	There were 30 more SSIs per 1000 participants with iodophor-alcohol surgical site skin preparation compared with chlorhexidine-alcohol	RR 0.614 (0.453 to 0.831)	2644 (3 studies)
Superficial SSI	47/1319 (3.6%)	73/1325 (5.5%)	There were 19 more superficial SSIs per 1000 participants with iodophor-alcohol surgical site skin preparation compared with chlorhexidine-alcohol	RR 0.653 (0.456 to 0.934)	2644 (3 studies)
Deep or Organ Space SSI	16/1319 (1.2%)	31/1325 (2.3%)	There were 11 more deep or organ space SSIs per 1000 participants with iodophor-alcohol surgical site skin preparation compared with chlorhexidine-alcohol	RR 0.522 (0.287 to 0.952)	2644 (3 studies)
Adverse events	2/572 (0.3%)	4/575 (0.7%)	There are 4 more adverse events per 1000 participants with iodophor-alcohol surgical site skin preparation compared with chlorhexidine-alcohol	RR 0.503 (0.092 to 2.733)	1147 (1 study)
**General surgery (including colorectal and abdominal)**					
All SSI	137/948 (14.5%)	158/911 (17.3%)	There were 28 more SSIs per 1000 participants with iodophor-alcohol surgical site skin preparation compared with chlorhexidine-alcohol	RR 0.844 (0.686 to 1.038)	1859 (3 studies)
Superficial SSI	49/495 (9.9%)	51/498 (10.2%)	There were 3 more superficial SSIs per 1000 participants with iodophor-alcohol surgical site skin preparation compared with chlorhexidine-alcohol	RR 0.968 (0.667 to 1.404)	993 (2 studies)
Deep or Organ Space SSI	44/495 (8.9%)	46/498 (9.2%)	There were 3 more deep or organ space SSIs per 1000 participants with iodophor-alcohol surgical site skin preparation compared with chlorhexidine-alcohol	RR 0.965 (0.653 to 1.427)	993 (2 studies)
Adverse events	0/392 (0.0%)	0/396 (0.0%)	There were no events reported	RR 1.010 (0.020 to 50.784)	788 (1 study)
**Skin and Soft Tissue Surgery**					
All SSI	2/204 (1.0%)	5/206 (2.4%)	There were 14 more SSIs per 1000 participants with iodophor-alcohol surgical site skin preparation compared with chlorhexidine-alcohol	RR 0.460 (0.105 to 2.024)	410 (2 studies)
Superficial SSI	2/204 (1.0%)	5/206 (2.4%)	There were 14 more superficial SSIs per 1000 participants with iodophor-alcohol surgical site skin preparation compared with chlorhexidine-alcohol	RR 0.460 (0.105 to 2.024)	410 (2 studies)
Deep or Organ Space SSI	0/204 (0.0%)	0/206 (0.0%)	There were no events reported	RR 1.012 (0.064 to 16.072)	410 (2 studies)
Adverse events	Not reported	Not reported	Not reported	Not estimable	No studies

95% CI: 95% Confidence interval; RR: Risk ratio. * Risk Ratio included continuity correction.

## Data Availability

Not applicable.

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
