# Peer review of "Randomised Controlled Trials of Alcohol-Based Surgical Site Skin Preparation for the Prevention of Surgical Site Infections: Systematic Review and Meta-Analysis"

_jcm, 2021, doi:10.3390/jcm10040663_

Round 1

Reviewer 1 Report

the reivew has been properly done, with lot's of dunkey work.

The authors are to be gratulated to identify difference among surgical procedures and disinfection.

CH X may be preferable for Ceasearn section - likely with a lot of protein load - but for all other types PVP iodine may be better. However, i do not understand the pathogenesis, but it is observed over and over. 

The high incidence in the Tuuli NEJM paper needs a point in the discussion, in particular because WHO change their recommendation based on this study.

8-vs 16 % is extremely high, and almost never observed in contintenal Europe.

points for improvement

the analysis requires stratification.

1. Since 1992, CDC critieria to define surgical site infections (SSIs) are standard of care. the authors need to present this subset of internatiolly recognized critiera for SSI

2. superficial SSIs are a nuissance, but do not change morbidity and mortality as well as length of hospital stay.

A subset analysis of deep and organ space infections is necessary before

Discussion: Studies did apply different concentration of iodine and CHX. Free iodine is the compound of antimicrobial effectiveness, and different concentrations will result in different incidence of SSIs.

Author Response

Thank you to the reviewer for their comments. We have provided a point-by-point response to each of the questions (in italics) raised by the reviewer.

1. Since 1992, CDC criteria to define surgical site infections (SSIs) are standard of care. the authors need to present this subset of internationally recognized criteria for SSI

As per the methods section, the primary outcome was SSI based on the CDC criteria, however, in the event, the trial did not apply the CDC definition, we mapped the applied definition to the CDC definition, where possible (line 83 - 87). 

Six studies applied the CDC definition (Veiga et al, Ngai et al, Tuuli et al, Broach et al, Kesani et al, Peel et al - see table 1). Running the meta-analysis including only those studies that applied the CDC definition, did not alter the results (RR 0.786, 95% 0.648 - 0.955). Given this was not a stated (a priori) part of the analysis plan and does not alter the results of the (requested) posthoc analysis, we have elected not to include this in the main manuscript. We are happy to be guided by the editor.

2. superficial SSIs are a nuisance but do not change morbidity and mortality as well as the length of hospital stay.

Superficial infections are associated with patient morbidity, drive antimicrobial use, and also are a risk fact for deep or organ/space infection. These infections form part of the CDC definition and are reported as part of the trial data. Therefore we have included this information in the meta-analysis sub-group analysis (table 2).

3. A subset analysis of deep and organ space infections is necessary 

This subgroup analysis has already been provided in table 2.

As per the stated analysis plan, the subgroups examined were superficial SSI and, deep or organ/space (line 125). Only three studies (Broach et al, Ngai et al and Rodrigues et al) specifically distinguished between deep and organ/space SSI.  Posthoc analysis, when including only those three studies, did not change the results  (RR1.20 95% CI 0.67-2.15: with very few events and very few studies). Given this was not a stated (a priori) part of the analysis plan and does not alter the results of the (requested) posthoc analysis, we have elected not to include this in the main manuscript. We are happy to be guided by the editor.

4. Studies did apply different concentrations of iodine and CHX. Free iodine is the compound of antimicrobial effectiveness, and different concentrations will result in the different incidence of SSIs.

We agree with the reviewer that a range of doses for the agents was utilized (lines 134-140) and highlighted this as a potential limitation (253 - 256). Given the lack of head to head trials comparing different concentrations, we are unable to draw any further conclusions.

Reviewer 2 Report

The paper is an interesting metanalysis comparing the use of alcohol-based chlorhexidine to alcohol-based iodophor preparations. Results suggest that alcohol-based chlorhexidine surgical site skin preparation is associated with a lower risk of surgical site infections for the cesarean section but with a higher risk for bone and joint surgical procedures, raising the possibility that the optimal skin preparation agent may differ with surgical procedures.

This metanalysis has two main limitations: the fact that studies do not report the exact concentration of the preparations and the limited number of trials.

Only some minor queries: I would report also in the conclusion the fact that, excluding the studies with the major risks of bias, iodophor preparation does not seem to have an advantage over alcohol-based iodophor preparations.

Thank You

Author Response

Thank you to the reviewer for their comments. 

We agree with the reviewer that there are a number of limitations to the analysis and that we advocate caution with the interpretation of the results. 

We have provided a point-by-point response to each of the questions (in italics) raised by the reviewer.

1. Only some minor queries: I would report also in the conclusion the fact that, excluding the studies with the major risks of bias, iodophor preparation does not seem to have an advantage over alcohol-based iodophor preparations.

We have amended the manuscript as suggested to include the following statement:

"However, when excluding studies at risk of major bias, this difference for bone and joint procedures was no longer significant. These observations must be interpreted with caution and require further investigation to corroborate these findings and to determine if there is a biological mechanism(s) explaining these findings." (Line 265 - 268)

We have also amended the abstract 

"However, when excluding studies at high risk of bias on sensitivity analysis, this difference in alcohol-based combinations for bone and joint surgery was no longer observed (RR 2.636; 95% CI 0.995, 6.983)....This difference must be interpreted with caution given the low number of studies and potential for bias, however, it warrants further investigation into the potential biological and clinical validity of these findings." (lines 28 - 30 and line 34).